published: 17 August 2021

# Effectiveness of a Video-Based Intervention on Reducing Perceptions of Fear, Loneliness, and Public Stigma Related to COVID-19: A Randomized Controlled Trial

Linda Valeri[1]*, Doron Amsalem[2], Samantha Jankowski[2], Ezra Susser[2,3] and Lisa Dixon[2]

[1]Department of Biostatistics, Columbia University, New York, NY, United States, [2]New York State Psychiatric Institute (NYSPI), New York, NY, United States, [3]Department of Epidemiology, Columbia University, New York, NY, United States

**Objectives:** During the first peak of the COVID-19 outbreak in the United States, we investigated the impact of digital interventions to reduce COVID-19 related fear, loneliness, and public stigma.

**Methods:** We recruited and randomly assigned 988 United States residents to: 1) no intervention 2) informational sheet to learn about COVID-19, 3) (2) AND video encouraging digital social activity, 4) (2) AND video sensitizing to COVID-19 related stigma (registered in Clinicaltrials.gov). Surveys were conducted between April 2-16, 2020. We employed generalized linear mixed models to investigate intervention effects.

**Results:** 10% of the participants reported not being afraid of people COVID-19+ and 32% reported not feeling lonely. Stigma and fear items reflected acute worries about the outbreak. Relative to the informational sheet only group, video groups led to greater reduction in perceptions of fear towards COVID-19+ (ORvideo.solo = 0.78, p-val<0.001; ORvideo.friend = 0.79, p-val<0.001) and of stigma (BETAvideo.solo = −0.50, p-val<0.001; BETAvideo.friend = −0.69, p-val<0.001).

**Conclusion:** Video-based interventions lead to reductions in COVID-19-related fear and stigma. No difference in social activity among groups was found, potentially explaining lack of efficacy on loneliness.

Keywords: COVID-19 mitigation measures, fear, loneliness, public stigma, randomized controlled trial, video intervention

**Edited by:**
*Robert Wellman,*
*University of Massachusetts Medical School, United States*

**Reviewed by:**
*Sherald Sanchez,*
*University of Toronto, Canada*

**\*Correspondence:**
*Linda Valeri*
*lv2424@columbia.edu*

This Original Article is part of the IJPH Special Issue "The impact of the COVID-19 pandemic on mental health"

**Citation:**
*Valeri L, Amsalem D, Jankowski S, Susser E and Dixon L (2021) Effectiveness of a Video-Based Intervention on Reducing Perceptions of Fear, Loneliness, and Public Stigma Related to COVID-19: A Randomized Controlled Trial. Int J Public Health 66:1604164. doi: 10.3389/ijph.2021.1604164*

## INTRODUCTION

The Coronavirus outbreak, COVID-19, which originated in mainland China in December 2019, spread rapidly to neighboring countries and Europe, and very soon thereafter to the Americas and other regions, was recognized as a pandemic on March 11th, 2020 by the World Health Organization. The outbreak has placed health care systems around the world under unprecedented stress, and the state of emergency has led non-essential businesses to shut down in most affected countries. Shelter-in-place orders (either enforced orders, unenforced orders, advisories), essential weapons to fight outbreaks similar to COVID-19, result in social isolation, influence the way individuals seek social interactions, and may enhance perceptions of fear, loneliness and public stigma towards individuals affected by COVID-19.

Public stigma refers to negative attitudes and beliefs that motivate individuals to fear, reject, avoid, and discriminate against other people. Stigma can emerge, as witnessed during the Ebola outbreak, when affected nations are labeled "infected countries" [1]. Uncertainty around the course of the epidemic, job insecurity, and social isolation can contribute to exacerbate perceptions of fear, loneliness and stigma, posing serious threats to individuals' mental health. In recent epidemics, isolation, loneliness, fear and stigma have precipitated depression, anxiety and post-traumatic stress disorder (PTSD) in both the general population and affected individuals [2, 3]. Researchers studying mental health effects of SARS found an association between duration of enforced quarantine and incidence of PTSD [4, 5].

Studies conducted during the first phase of the COVID-19 pandemic have identified moderate to high levels of loneliness [6, 7] and found associations between changes in loneliness status and depression [8, 9]. Since the start of the COVID-19 pandemic, evidence was also gathered on stigma belief and stigma messages along with misinformation and conspiracy theories [10–12]. Stigma messages were particularly reported towards healthcare workers and COVID-19 positive (+) individuals [13].

Despite the wealth of recent research on the impacts of COVID-19 on mental health, gaps remain, as most of these studies are cross-sectional and few of them directly address fear, stigma and loneliness. Moreover, to our knowledge, studies that have examined the efficacy of an intervention in reducing COVID-19 related fear, loneliness and public stigma are very limited [14–16].

In a time in which individuals all over the world are experiencing the spread of a pandemic, it is critical to evaluate the potential surge of the phenomena of fear, public stigma, and of perceived loneliness, as these can impact behavior [4], mental and physical health. Fischer et al. (2019) [3], summarized intervention studies targeting public communicable disease-related stigma and social anxiety conducted in real-world settings. Providing clear information about the outbreak, involving community leaders in anti-stigma campaigns early in the outbreak, and encouraging social contact (in medically accepted fashion) are the most effective interventions to improve quality of life [2, 17]. Moreover, interventions aiming at promoting mental health would support adherence to isolation practices [18]. It is therefore critical not only to monitor COVID-19 related fear, loneliness and public stigma but also to provide effective strategies to intervene early to tame them and to prevent the downstream mental health consequences.

Video interventions have been shown to be effective in mitigating mental health stigma [19, 20]. However, to our knowledge, Evidence of the effectiveness of video-based interventions to mitigate COVID-19 related fear, loneliness and stigma is very limited [14–16]. Video interventions are a feasible option during isolation, and are in addition, easy and cost-effective to implement. Therefore, we designed two video interventions following recent guidelines suggesting employing knowledge-shaping and attitude-changing content to reduce fear, stigma and discrimination associated with COVID-19 [14, 15] and to promote and facilitate social interactions to reduce loneliness perceptions [16].

Our study aims to accomplish two goals. First, we describe COVID-19 related fear, public stigma and loneliness perceptions between April 2nd, 2020 and April 16th, 2020, while the COVID-19 outbreak was spreading and peaking for the first time among the US population, and most states had executed shelter-in-place orders. Second, we evaluate the effect of two video-based interventions that we deployed during this time period with the goal of reducing perceptions of fear, loneliness, and public stigma. In the context of uncertainty and rapid changes at the early phase of the pandemic, when multiple unknown factors may influence stigma, fear and loneliness perception, the RCT design allows to evaluate rigorously the efficacy of prevention measures. We compare the video interventions to a control group and an informational sheet providing basic information and recommendations to prevent the spread of COVID-19.

## METHODS

### Overview of Study Design and Study Population

This is a 2-week longitudinal study with data collection at four time points during April 2020. Data were collected at baseline pre-intervention (April 2), immediate post-intervention (April 2) and at two subsequent time points 1 week apart (April 9, follow-up 1, and April 16, follow-up 2). The study was approved by the authors' institutions review boards and registered in Clinicaltrials.gov. Participants responded to survey questionnaires at each time (**Supplementary Appendix S1**). Adult residents of the United States between the age of 18 and 70 were eligible for the study. We recruited participants using Amazon Mechanical Turk (AMT) [21, 22]. A posting was listed on AMT explaining the terms and conditions of the study. Participants who met the eligibility criteria and agreed to the conditions and consent to participate, completed the study procedures *via* Qualtrics.com, a secure online data-collection platform (conducted by co-author DA).

### Intervention

Participants were randomly assigned to one of four intervention arms: 1) control arm (no intervention) 2) a written informational sheet developed by European Centre for Disease Prevention and Control to learn about the origin of the COVID-19 outbreak and to educate the public on the best prevention strategies (**Supplementary Appendix S2**), 3) the informational sheet and a 150-s video representing a video call among two friends sheltering-in-place ("video.friend", **Supplementary Appendix S3**), and 4) the informational sheet and a 90-s video representing a COVID-19 patient ("video.solo", **Supplementary Appendix S3**).

The "video.friend" included a video conversation between two friends, their concerns about older family members, and the impact of COVID-19 and/or quarantine on their perceptions of loneliness and fear. As previous research found that social facilitation interventions may improve social support and reduce loneliness perceptions [16, 23], the "video.friend" was designed to encourage the engagement in social interactions using the safe

videoconference avenue and to share the experience of living through the pandemic with friends. The "video.solo" included a personal introduction of a COVID-19 patient, description of recent events in relation to covid-19, description of how the environment reacted, the negative feelings it evoked and concerns about being stigmatized. The "video.solo" was designed to address perceptions of stigma and fear associated with COVID-19 by increasing awareness towards the disease and the emotional connection with COVID-19 patients. We hypothesized that both video-based intervention groups would demonstrate lower rates of fear than the informational sheet and control groups. We additionally hypothesized that the "video.solo" group would have lower rates of stigma and that the "video.friend" would demonstrate lower rates of loneliness than the informational sheet and control groups.

## Measures

Baseline questionnaires included questions on demographic information (age, gender, race), a stigma instrument and questions about COVID-19 outbreak-related fear and loneliness (**Supplementary Appendix S1**). The stigma instrument included three items that suggested that people with COVID-19 could feel: 1) guilty, 2) need to hide, and 3) avoid friends because of COVID-19. The items were modified to COVID-19 from a stigma toward HIV questionnaire [24].

Follow-up questionnaires, in addition to the stigma, fear and loneliness instruments, included questions about quarantine status, internet use, and social contact seeking behavior.

Primary outcomes include *loneliness* (How lonely do you feel? Ordinal variable), *general fear score* (continuous, score is obtained summing the items: How much are you afraid to be isolated because of COVID-19?, How much are you afraid to be diagnosed with COVID-19?, How much are you afraid of the consequences of the COVID-19 outbreak?), public *stigma score* (continuous, score is obtained summing the items: It's easier to avoid friends than worry about telling someone about having COVID-19, People feel guilty because they have COVID-19, I worry that people may judge me if I had COVID-19, People with COVID-19 never feel the need to hide the fact that they have COVID-19), *stigma-related fear* (Are you afraid of people COVID-19 positive (+)? ordinal variable). We considered the "*stigma-related fear*" outcome separately from the *general fear score* because it combines both fear and stigma perceptions. We considered public stigma items as we expected to enroll none or very limited participants affected by COVID-19. We considered the item fear of people COVID-19 + separately from the items included in the fear score, as it might capture stigmatization perceptions along with fear for the contagion.

Potential mechanisms explaining the intervention effects may involve changes in digital activity or social engagement. Therefore, we considered time spent on the internet (for leisure or work-related activities) in the past week (<1 h, 1h–4h, 4h–7h, >7 h) and social contact seeking behavior in the past week (How many times did you contact [in person or via video call] your friends/family members over the last week? 0–2, 2–5, 5–7, more than 7) as secondary outcomes. Primary outcomes were measured at baseline pre- and post- intervention, and at two follow-up points 1 week apart. Secondary outcomes were measured at baseline and at two follow-up points.

## Statistical Analysis

We provided descriptive statistics for the outcomes and potentially relevant baseline predictors a priori specified (gender, race, age, quarantine status, living in high vs. low risk states during first peak of the pandemic in the United States, time spent on internet, and number of social contacts sought) by intervention groups.

We then evaluated the effect of the intervention on trends in the primary outcomes using generalized linear mixed models (GLMM). We used a linear link function to model *stigma* and *general fear scores* and ordinal logistic link for *loneliness* and *stigma related fear* items. All longitudinal regression models included a time factor, intervention group, and their interaction. Models were adjusted for baseline scores and baseline covariates to minimize confounding bias. In particular, we planned for these comparisons: "video.solo" vs. informational sheet, "video.friend" vs. informational sheet.

A power calculation assuming a sample size of 988, 75% retention rate along the four time points of data collection (0.5 within correlation), and an alpha level of 0.001, indicated that our study was powered at the 80% level to detect a difference in means of the continuous outcomes (stigma and general fear score, mean = 10 in control group and sd = 2.5 equal across groups) of 0.66 units comparing intervention groups to informational sheet/control group.

Finally, we investigated the effect of the intervention over the follow-up on the secondary outcomes, time spent on internet and social contact seeking behavior in the past week, and the association between primary outcomes at baseline and the secondary outcomes using mixed ordinal logistic regression.

All tests were two-tailed, we used a significance level threshold of 0.001 to account for multiple testing. Statistical analysis was performed on R studio version 1.2.5042. As sensitivity analyses, we replicated intervention effect analyses using inverse probability of censoring weights (IPCW) for drop-out. The IPCW analysis considered baseline covariates, baseline outcome scores and intervention groups as predictors of dropout. Weights were truncated at the 95th percentile to tame the issue of extreme weights.

## RESULTS

### Participants Characteristics at Baseline

The study included 988 participants from across 49 states in the US. Except Nebraska, South Dakota, North Dakota, Iowa and Arkansas, all states had a state-wide or a partial shelter-in-place order enacted during the follow-up of our study. A total of 29.7% of the participants were residents of states with high risk of COVID-19 as of April 1, 2020 (WA/CA/NY/NJ/MI/IL/LA), 59% were male, 77% identified as White, 9% as African American, 9% as Asian, and 3% as Native American and the average age was 37.1 (SD = 11.9). At baseline 17.6% of the participants reported

**TABLE 1** | Baseline characteristics by intervention arm.

| | Control (N = 250) | Informational sheet (N = 243) | Video.Solo (N = 249) | Video.Friend (N = 246) | Total (N = 988) |
|---|---|---|---|---|---|
| Age (years) | | | | | |
| Mean (SD) | 36.0 (10.6) | 38.1 (12.6) | 37.0 (11.8) | 37.4 (12.4) | 37.1 (11.9) |
| Median [Min, Max] | 34.0 [18.0, 70.0] | 34.0 [18.0, 70.0] | 35.0 [19.0, 69.0] | 33.0 [19.0, 70.0] | 34.0 [18.0, 70.0] |
| COVID-19 State Risk | | | | | |
| High Risk | 67 (26.8%) | 70 (28.8%) | 88 (35.3%) | 68 (27.6%) | 293 (29.7%) |
| Low Risk | 183 (73.2%) | 173 (71.2%) | 161 (64.7%) | 178 (72.4%) | 695 (70.3%) |
| Gender | | | | | |
| Male | 144 (57.6%) | 154 (63.4%) | 156 (62.7%) | 136 (55.3%) | 590 (59.7%) |
| Female | 106 (42.4%) | 84 (34.6%) | 93 (37.3%) | 107 (43.5%) | 390 (39.5%) |
| Transgender | 0 (0%) | 1 (0.4%) | 0 (0%) | 1 (0.4%) | 2 (0.2%) |
| Other | 0 (0%) | 0 (0%) | 0 (0%) | 2 (0.8%) | 2 (0.2%) |
| Missing | 0 (0%) | 4 (1.6%) | 0 (0%) | 0 (0%) | 4 (0.4%) |
| Race/Ethnicity | | | | | |
| African American | 19 (7.6%) | 26 (10.7%) | 26 (10.4%) | 17 (6.9%) | 88 (8.9%) |
| Asian | 21 (8.4%) | 21 (8.6%) | 15 (6.0%) | 31 (12.6%) | 88 (8.9%) |
| Native American | 4 (1.6%) | 4 (1.6%) | 3 (1.2%) | 6 (2.4%) | 17 (1.7%) |
| White | 201 (80.4%) | 182 (74.9%) | 190 (76.3%) | 189 (76.8%) | 762 (77.1%) |
| Prefer not to answer | 0 (0%) | 4 (1.6%) | 3 (1.2%) | 2 (0.8%) | 9 (0.9%) |
| Other | 5 (2.0%) | 6 (2.5%) | 12 (4.8%) | 1 (0.4%) | 24 (2.4%) |
| Social contacts past week | | | | | |
| 0–2 | 54 (21.6%) | 59 (24.3%) | 54 (21.7%) | 63 (25.6%) | 230 (23.3%) |
| 2–5 | 98 (39.2%) | 86 (35.4%) | 84 (33.7%) | 82 (33.3%) | 350 (35.4%) |
| 5–7 | 35 (14.0%) | 43 (17.7%) | 37 (14.9%) | 43 (17.5%) | 158 (16.0%) |
| More than 7 | 62 (24.8%) | 54 (22.2%) | 74 (29.7%) | 57 (23.2%) | 247 (25.0%) |
| Missing | 1 (0.4%) | 1 (0.4%) | 0 (0%) | 1 (0.4%) | 3 (0.3%) |
| Time on internet past week | | | | | |
| Less than 1 hour | 1 (0.4%) | 6 (2.5%) | 2 (0.8%) | 2 (0.8%) | 11 (1.1%) |
| 1–4 h | 49 (19.6%) | 51 (21.0%) | 53 (21.3%) | 48 (19.5%) | 201 (20.3%) |
| 4–7 h | 90 (36.0%) | 95 (39.1%) | 100 (40.2%) | 88 (35.8%) | 373 (37.8%) |
| More than 7 h | 110 (44.0%) | 91 (37.4%) | 94 (37.8%) | 108 (43.9%) | 403 (40.8%) |
| Quarantine status | | | | | |
| No | 151 (60.4%) | 149 (61.3%) | 146 (58.6%) | 150 (61.0%) | 596 (60.3%) |
| Yes–me | 47 (18.8%) | 45 (18.5%) | 42 (16.9%) | 40 (16.3%) | 174 (17.6%) |
| Yes–friend/family member | 49 (19.6%) | 49 (20.2%) | 61 (24.5%) | 53 (21.5%) | 212 (21.5%) |
| Missing | 3 (1.2%) | 0 (0%) | 0 (0%) | 3 (1.2%) | 6 (0.6%) |

being quarantined and 25% reported of knowing a friend or a family member quarantined. At baseline 3% of the participants reported being COVID-19+. These baseline characteristics were approximately balanced across intervention arms (**Table 1**). Baseline scores for stigma, fear and loneliness items were high. Only 10% reported not being afraid of people COVID-19+ and 32% of the participants reported not feeling lonely.

All stigma and fear items reflected acute worries about the outbreak. At baseline participants reporting to be quarantining or knowing someone in quarantine displayed higher *general fear score* (beta$_{quarantine-me}$ = 0.66, p-val = 0.004; OR$_{quarantine-other}$ = 0.84, $p$ < 0.0001), *stigma score* (beta$_{quarantine-me}$ = 0.93, p-val<0.0001; OR$_{quarantine-other}$ = 0.36, $p$ = 0.06), *fear of COVID-19+ patients* (oOR$_{quarantine-me}$ = 1.40, p-val = 0.26; OR$_{quarantine-other}$ = 1.87, $p$ = 0.004), and *loneliness* perceptions (OR$_{quarantine-me}$ = 1.85, p-val = 0.001; OR$_{quarantine-other}$ = 1.55, $p$ = 0.01). Some imbalances due to chance across intervention arms according to baseline outcomes were observed (**Table 2**). Out of the 988 participants 25% were lost to follow-up. Participants missing at follow-up displayed higher baseline scores and were more likely to be quarantined then participants who stayed in the study (**Supplementary Table S1**).

## Intervention Effects on Primary Outcomes
### Informational Sheet Compared to Control Group

We found no difference in primary or secondary outcomes between those randomized to the informational sheet compared to those randomized to the control condition (**Figure 1**). This was expected because none of the content of the document was aiming at reducing COVID-19 related stigma, loneliness, or fear.

## Video Interventions Compared to Informational Sheet

Video interventions displayed effects on the primary outcomes (**Figure 1**). Relative to the group assigned to the informational sheet, both video groups led to a reduction in stigma perceptions relative to the vignette group during the follow-up, with video. friend displaying a more persistent reduction over time (beta$_{video.solo}$ = −0.92, CI = −1.28−−0.55; beta$_{time*video.solo}$ = 0.21, CI = 0.04–0.38; beta$_{video. friend}$ = −0.85, CI = −1.20−−0.44; beta$_{time*video.friend}$ = 0.08, CI = −0.08–0.25, **Table 3**). Compared to the information sheet group, participants assigned to the video arms displayed a more marked reduction in the stigma related fear outcome (OR$_{time*video.solo}$ = 0.61, CI = 0.50–0.75;

**TABLE 2** | Baseline levels of primary outcomes by intervention arm.

| | Control (N = 250) | Informational sheet (N = 243) | Video.Solo (N = 249) | Video.Friend (N = 246) | Total (N = 988) |
|---|---|---|---|---|---|
| **Loneliness** | | | | | |
| Not at all | 84 (33.6%) | 81 (33.3%) | 72 (28.9%) | 83 (33.7%) | 320 (32.4%) |
| A bit | 72 (28.8%) | 73 (30.0%) | 86 (34.5%) | 92 (37.4%) | 323 (32.7%) |
| Quite a bit | 55 (22.0%) | 59 (24.3%) | 48 (19.3%) | 38 (15.4%) | 200 (20.2%) |
| A lot | 37 (14.8%) | 30 (12.3%) | 40 (16.1%) | 32 (13.0%) | 139 (14.1%) |
| Missing | 2 (0.8%) | 0 (0%) | 3 (1.2%) | 1 (0.4%) | 6 (0.6%) |
| **Avoid friends when Covid +** | | | | | |
| Strongly disagree | 20 (8.0%) | 16 (6.6%) | 28 (11.2%) | 19 (7.7%) | 83 (8.4%) |
| Disagree | 44 (17.6%) | 52 (21.4%) | 54 (21.7%) | 55 (22.4%) | 205 (20.7%) |
| Agree | 104 (41.6%) | 109 (44.9%) | 95 (38.2%) | 98 (39.8%) | 406 (41.1%) |
| Strongly agree | 77 (30.8%) | 59 (24.3%) | 66 (26.5%) | 66 (26.8%) | 268 (27.1%) |
| Missing | 5 (2.0%) | 7 (2.9%) | 6 (2.4%) | 8 (3.3%) | 26 (2.6%) |
| **Guilt when Covid +** | | | | | |
| Strongly disagree | 18 (7.2%) | 18 (7.4%) | 27 (10.8%) | 29 (11.8%) | 92 (9.3%) |
| Disagree | 94 (37.6%) | 70 (28.8%) | 74 (29.7%) | 76 (30.9%) | 314 (31.8%) |
| Agree | 94 (37.6%) | 117 (48.1%) | 106 (42.6%) | 101 (41.1%) | 418 (42.3%) |
| Strongly agree | 38 (15.2%) | 32 (13.2%) | 38 (15.3%) | 33 (13.4%) | 141 (14.3%) |
| Missing | 6 (2.4%) | 6 (2.5%) | 4 (1.6%) | 7 (2.8%) | 23 (2.3%) |
| **Judged when Covid +** | | | | | |
| Strongly disagree | 47 (18.8%) | 34 (14.0%) | 55 (22.1%) | 44 (17.9%) | 180 (18.2%) |
| Disagree | 65 (26.0%) | 77 (31.7%) | 66 (26.5%) | 72 (29.3%) | 280 (28.3%) |
| Agree | 84 (33.6%) | 86 (35.4%) | 75 (30.1%) | 75 (30.5%) | 320 (32.4%) |
| Strongly agree | 50 (20.0%) | 41 (16.9%) | 47 (18.9%) | 50 (20.3%) | 188 (19.0%) |
| Missing | 4 (1.6%) | 5 (2.1%) | 6 (2.4%) | 5 (2.0%) | 20 (2.0%) |
| **No need to hide when Covid +** | | | | | |
| Strongly disagree | 47 (18.8%) | 34 (14.0%) | 55 (22.1%) | 44 (17.9%) | 180 (18.2%) |
| Disagree | 65 (26.0%) | 77 (31.7%) | 66 (26.5%) | 72 (29.3%) | 280 (28.3%) |
| Agree | 84 (33.6%) | 86 (35.4%) | 75 (30.1%) | 75 (30.5%) | 320 (32.4%) |
| Strongly agree | 50 (20.0%) | 41 (16.9%) | 47 (18.9%) | 50 (20.3%) | 188 (19.0%) |
| Missing | 4 (1.6%) | 5 (2.1%) | 6 (2.4%) | 5 (2.0%) | 20 (2.0%) |
| **Fear of people Covid +** | | | | | |
| Not at all | 28 (11.2%) | 13 (5.3%) | 34 (13.7%) | 25 (10.2%) | 100 (10.1%) |
| A bit | 59 (23.6%) | 70 (28.8%) | 52 (20.9%) | 58 (23.6%) | 239 (24.2%) |
| Quite a bit | 84 (33.6%) | 75 (30.9%) | 72 (28.9%) | 82 (33.3%) | 313 (31.7%) |
| A lot | 78 (31.2%) | 82 (33.7%) | 89 (35.7%) | 79 (32.1%) | 328 (33.2%) |
| Missing | 1 (0.4%) | 3 (1.2%) | 2 (0.8%) | 2 (0.8%) | 8 (0.8%) |
| **Fear of isolation** | | | | | |
| Not at all | 78 (31.2%) | 93 (38.3%) | 91 (36.5%) | 84 (34.1%) | 346 (35.0%) |
| A bit | 67 (26.8%) | 70 (28.8%) | 66 (26.5%) | 60 (24.4%) | 263 (26.6%) |
| Quite a bit | 51 (20.4%) | 45 (18.5%) | 48 (19.3%) | 56 (22.8%) | 200 (20.2%) |
| A lot | 51 (20.4%) | 35 (14.4%) | 42 (16.9%) | 44 (17.9%) | 172 (17.4%) |
| Missing | 3 (1.2%) | 0 (0%) | 2 (0.8%) | 2 (0.8%) | 7 (0.7%) |
| **Fear of Covid + diagnosis** | | | | | |
| Not at all | 24 (9.6%) | 20 (8.2%) | 32 (12.9%) | 21 (8.5%) | 97 (9.8%) |
| A bit | 85 (34.0%) | 85 (35.0%) | 74 (29.7%) | 73 (29.7%) | 317 (32.1%) |
| Quite a bit | 66 (26.4%) | 74 (30.5%) | 74 (29.7%) | 77 (31.3%) | 291 (29.5%) |
| A lot | 75 (30.0%) | 64 (26.3%) | 68 (27.3%) | 75 (30.5%) | 282 (28.5%) |
| Missing | 0 (0%) | 0 (0%) | 1 (0.4%) | 0 (0%) | 1 (0.1%) |
| **Fear of outbreak consequences** | | | | | |
| Not at all | 9 (3.6%) | 5 (2.1%) | 17 (6.8%) | 8 (3.3%) | 39 (3.9%) |
| A bit | 42 (16.8%) | 60 (24.7%) | 35 (14.1%) | 49 (19.9%) | 186 (18.8%) |
| Quite a bit | 85 (34.0%) | 72 (29.6%) | 81 (32.5%) | 81 (32.9%) | 319 (32.3%) |
| A lot | 114 (45.6%) | 106 (43.6%) | 116 (46.6%) | 108 (43.9%) | 444 (44.9%) |

$OR_{time*video.friend} = 0.71$, CI = 0.58–0.86, **Table 3**). No effects of the video interventions on loneliness nor on the general fear score were found. A weighted regression analysis adjusting for loss to follow-up using stabilized inverse probability of censoring weights (**Supplementary Tables S2, S3**) confirmed the effects of the video interventions.

## Secondary Analyses

No statistically significant differences meeting our multiple comparison $p$-value<0.001 between the video intervention groups to control and informational sheet group in time spent on internet and in self-reported social contacts were detected (**Supplementary Figure S1** and **Supplementary Table S4**).

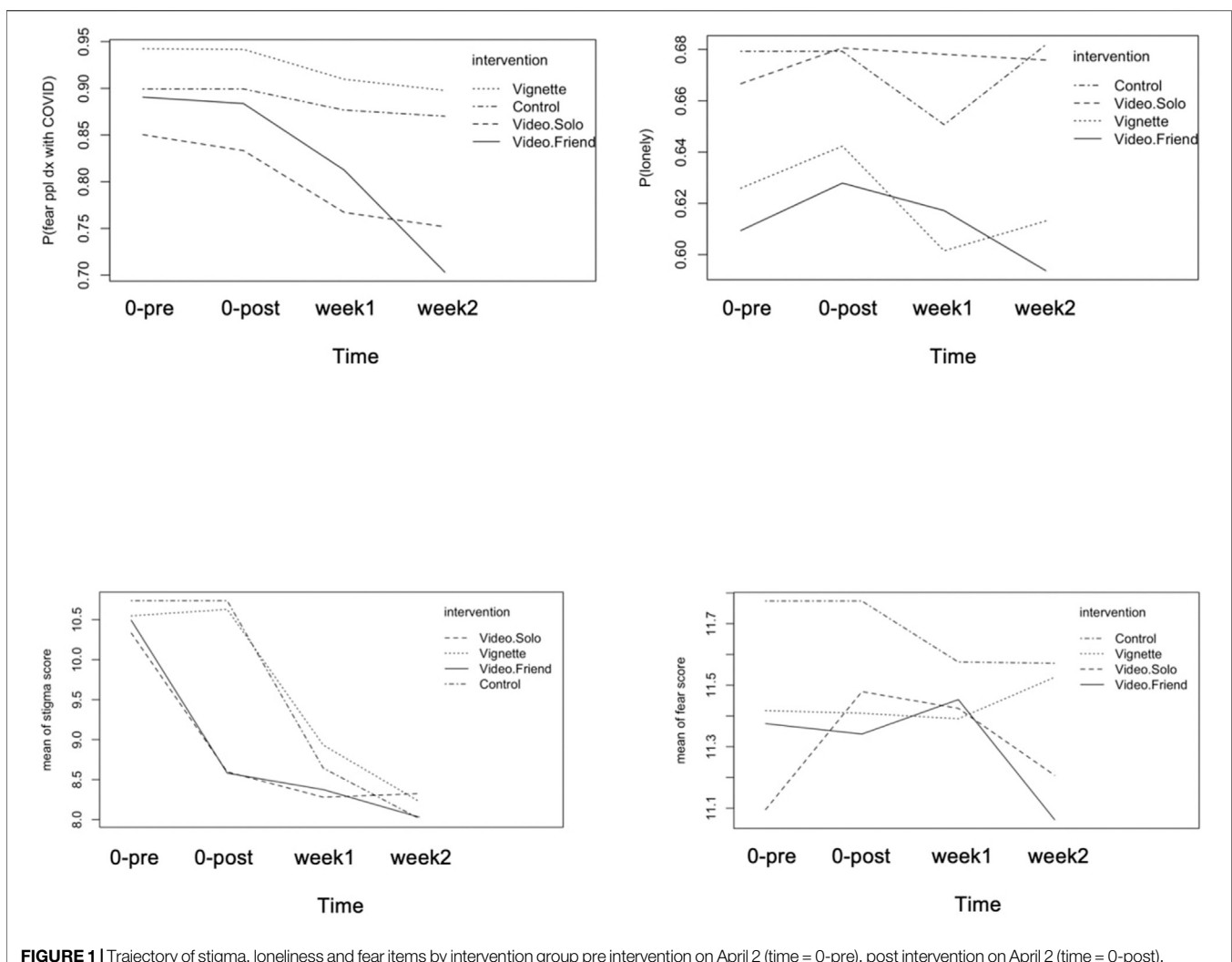

**FIGURE 1 |** Trajectory of stigma, loneliness and fear items by intervention group pre intervention on April 2 (time = 0-pre), post intervention on April 2 (time = 0-post), follow-up 1 on April 9 (week 1), follow-up 2 on April 16 (week 2).

**TABLE 3 |** Longitudinal intervention effects comparing video groups to informational sheet group on primary outcomes adjusting for covariates (age, quarantine status, gender, risk, race, baseline score).

| Predictors | Total stigma score | Total fear score | Fear of people Covid + | Loneliness |
|---|---|---|---|---|
|  | Difference in means | Difference in means | Odds Ratios | Odds Ratios |
| Time | −1.02 *** (−1.13–−0.91) | −0.09 (−0.18–0.00) | 0.83** (0.73–0.94) | 1.00 (0.74–1.37) |
| Video.Solo vs. informational sheet | −0.92 *** (−1.28–−0.55) | 0.11 (−0.17–0.40) | 1.54 (0.76 –3.01) | 1.05 (0.44–2.05) |
| Video.Friend vs. informational sheet | −0.85 *** (−1.20–−0.44) | 0.00 (−0.29–0.30) | 1.15 (0.57–2.32) | 0.46 (0.17–1.21) |
| Time* [Video.Solo vs. informational sheet] | 0.21* (0.04–0.38) | 0.01 (−0.12–0.14) | 0.61*** (0.50–0.75) | 1.02 (0.81–1.29) |
| Time* [Video.Friend vs. informational sheet] | 0.08 (−0.08–0.25) | −0.09 (−0.23–0.04) | 0.71*** (0.58–0.86) | 1.27 (0.98–1.64) |

*p<0.05, **p<0.01, ***p<0.001.

As individuals reporting of quarantining or knowing someone quarantining at baseline displayed higher fear, stigma and loneliness perceptions, in a secondary analysis we evaluated whether the intervention effects on loneliness, fear and stigma perceptions were modified by quarantine status. We did not find evidence of such an effect modification, however our RCT was planned and powered to detect main effects and underpowered to detect interactions of clinically relevant magnitude.

# DISCUSSION

We evaluated the impact of video-based interventions on the reduction of COVID-19 related fear, loneliness, and public stigma attitudes among the US general population during April 2–16th, 2020. The study was conducted during the surge of the pandemic, as the highest number of new cases per day in the United States was 43,438 on April 6th. Recently contributions have been

published highlighting COVID-19 related loneliness and its implications for mental health [6–9].

We observed high levels of fear, loneliness and presence of public stigma attitudes during the assessment period. The high level of fear, loneliness and stigma confirmed what has been seen in other epidemics [1, 25].

Encouraging reductions over time in the stigma score and the item "fear towards COVID-19 patients", which is related to public stigma, were observed. Individuals with COVID-19 may anticipate prejudice and aggression ("people might think that it was my fault that, I got infected. No one will want to meet or speak with me. I am a dangerous person") or internalize public stereotypes of people with infectious illness ("It is my fault. If I would wash my hands more often, it wouldn't happen"). The misinformation and politicization of this pandemic contributes to stigma as well. Many people consume false narratives, mostly on social media, leading to the misconception of numerous public health guidelines.

This is important because as a consequence of the fear of being stigmatized, individuals in the general population may avoid testing themselves, and individuals who tested positive may hesitate to tell others. Long lasting negative thoughts associated with fear, isolation and public stigma, may lead to anxiety and depression even among individuals who are not COVID-19+. In sum, public stigma induces shame and fear, may decrease healthcare seeking, and create barriers to treatment. This underscores the importance of identifying effective strategies that can change perceptions of stigma and of fear towards COVID-19 patients.

We hypothesize that participants' identification with the video presenters (a COVID-19 patient, and two friends supporting each other during the shelter-in-place) may explain the video-based interventions effect in reducing stigma score and stigma-related fear [26, 27]. We did not find evidence of a behavioral pathway operating as no differences in self-reported social contacts and internet use was found among the intervention groups. Intervention effects on the stigma items appeared stronger in the short term and reduced over time, while intervention effects on fear of COVID-19+ were sustained. This could be due to the ongoing exposure to COVID-19 information and to testimonials from COVID-19 patients, which may have led to reductions in stigma scores over time for all participants, and leading control and informational sheet groups to catch up with the video groups.

We found that during the 2 weeks of follow-up the levels of general fear score and loneliness remained high, as the first peak of the pandemic was unfolding. Our interventions did not reduce the general fear score and loneliness perceptions, which remained high during the surge of the outbreak. It is unlikely that duration of videos significantly contributed to lack of efficacy, as there is a growing body of literature showing that short videos can be as effective as long ones [19, 28, 29]. Dosage or intensity of the video-intervention might be a determining factor. Other studies showed the benefit of adding a booster (showing the video more than once) [30]. Failure to show significant reductions in fear and loneliness could be also attributed to inadequate content of the video-based interventions. We initially hypothesized that "video.friend" could have led to reductions in loneliness and fear by encouraging digital social interactions based on previous studies [31] that showed that

intentions may lead to a change in behavior. However, no difference was found in self-reported social contacts across groups during the follow-up. A follow-up of 2 weeks might be too short to establish meaningful social contacts and to observe changes in loneliness perceptions. Others have noted that feeling of loneliness is hard to modify in such a short span of time [32, 33]. However, a change in perception does not always lead to a change in behavior. Finally, the interventions were deployed in the time of acute manifestations of these perceptions due to uncertainties around the epidemic; hospitalizations and death rates were increasing in some states at this time through the month of April. Taken together these facts could explain the lack of efficacy.

Our study suggests directions for research. Future interventions should explore whether tailoring the video content according to the audience characteristics such as age, gender, and race could deliver larger effects. Interventional studies that target reducing fear and loneliness during an epidemic are warranted. In particular, effectively facilitating behavior change in addition to perceptions could be key to reduce loneliness. It is important to identify social activities mediated through digital devices that can promote the development of stable social relationships in times of epidemic related shelter-in-place. Furthermore, it is crucial to outline a framework for integrating social media as a tool in managing the current evolving pandemic, as directing people to trusted sources, fight misinformation, and use social media as a diagnostic tool and referral system [22]. A qualitative study would be needed to improve the design and content of the video-interventions.

Our study has several strengths. We introduced inexpensive video-based interventions to reduce stigma, fear and loneliness attitudes during the COVID-19 pandemic. The longitudinal design allowed us to monitor trends in primary and secondary outcomes and to assess the intervention effects and potential mechanism of action of the interventions. The timing of the study, geographical spread and wide age range of the participants, allowed us to capture the US population during the surge of the outbreak. Our study had a good retention rate.

Some limitations in our study should also be noted. We employed AMT to recruit participants and AMT workers, a population of heavy internet users [34]. It is possible that the individuals most affected by the COVID-19 pandemic were the least likely to participate in the study. Caution should be placed in transporting our findings to the general US population due to potential selection bias [35]. To maximize retention rate, we designed a brief questionnaire. While the stigma questions were adapted for the purposes of this study from a validated instrument, loneliness and fear questions were not. This could lead to measurement error in our primary outcomes. Measurement error in self-reported secondary outcomes could have also reduced our power to detect intervention effects. The follow-up of our study is 2 weeks, due to resource constraints and to maximize retention rate. Therefore, we were unable to provide evidence of the long-term effects of the video-based interventions.

Our study did not include instruments to capture psychopathology such as depression and anxiety. This limited our ability to assess the relationship between COVID-19 related fear, stigma and loneliness as well as participants social activity

and internet use with long term mental health outcomes. Further research should include broad assessment of mental health clinical outcomes as well.

Despite these limitations, we were able to capture longitudinally a sample of the US population during the first peak of the pandemic. This is clinically meaningful and unique, as no studies were able to provide a randomized intervention combined with during the early stages of the pandemic. We showed that video interventions are effective and easily disseminated. Changing early stigma and fear perceptions have the potential of reducing the epidemic toll on mental health as well as increasing compliance to shelter-in-place and other disease containment strategies.

## Public Health Implications

This longitudinal intervention study provides evidence on the high levels of COVID-19 related fear, loneliness and stigma attitudes, underscoring the importance of continuous monitoring and of taking early actions to minimize the long-term consequences that these perceptions may have on the health of the individuals and the society at large. Video-based interventions aiming at sensitizing individuals to the consequences of the outbreak and encouraging digital social interactions are feasible and effective in mitigating fear and stigma during the acute phases of the pandemic.

## DATA AVAILABILITY STATEMENT

The datasets generated during and analyzed during the current study are available from the corresponding author on reasonable request.

## ETHICS STATEMENT

The studies involving human participants were reviewed and approved by New York Psychiatric Institute and Columbia

University Institutional review boards and registered in Clinicaltrials.gov (identifier: NCT04734171). The patients/participants provided their written informed consent to participate in this study.

## AUTHOR CONTRIBUTIONS

LV conceptualized the study, conducted statistical analysis, interpreted results, drafted and revised manuscript receiving feedback from co-authors DA conceptualized the study, lead data collection, interpreted results, reviewed manuscript SJ interpreted results, reviewed manuscript ES interpreted results, reviewed manuscript LD conceptualized the study, interpreted results, reviewed manuscript.

## FUNDING

This work was supported by the National Institute of Mental Health award K01 MH118477 and the Columbia Innovation Award.

## CONFLICT OF INTEREST

The authors declare that the research was conducted in the absence of any commercial or financial relationships that could be construed as a potential conflict of interest.

## SUPPLEMENTARY MATERIAL

The Supplementary Material for this article can be found online at: https://www.ssph-journal.org/articles/10.3389/ijph.2021.1604164/full#supplementary-material

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
