## [Reviewer comments · International Journal of Public Health]

Peer Review Report

Review Report on Effectiveness of a video-based intervention on reducing perceptions of fear, loneliness, and public stigma related to COVID-19: a Randomized Controlled Trial

Original Article, Int J Public Health

Reviewer: Sherald Sanchez

Submitted on: 26 May 2021

Article DOI: 10.3389/ijph.2021.1604164

EVALUATION

Q 1 Please provide your detailed review report to the authors. The editors prefer to receive your review structured in major and minor comments. Please consider in your review the methods (statistical methods valid and correctly applied (e.g. sample size, choice of test), is the study replicable based on the method description?), results, data interpretation and references. If there are any objective errors, or if the conclusions are not supported, you should detail your concerns.

Congratulations to the authors for completing this study and manuscript and thank you for the opportunity to provide feedback on this piece of scholarly work. Given my area of research, I've refrained from commenting on the statistical methods. Instead, I focused on areas in which I have significant experience, including the development of of complex interventions in mental health, behavioural and psychosocial interventions, and digital health design. My comments, both major and minor, are attached.

Q 2 Please summarize the main findings of the study.

The use of an informative video showing an individual who is isolating due to COVID-19 led to a reduction in perceptions of fear and stigma towards COVID-19 and people with COVID-19. On the other hand, the use of an informative video showing two people on a video call chatting about COVID-19 had no effect on perceived loneliness.

Q 3 Please highlight the limitations and strengths.

Strengths: The methods and results pertaining to perceptions of fear and stigma towards individuals with COVID-19, as reported in the manuscript, are markedly stronger compared to the findings on loneliness and social activity. The RCT appears well-designed and the results, including the analysis, meticulously reported. Limitations include issues on appropriateness of RCT and secondary outcome measures relative to the stated objectives and lack of clarity in describing the interventions (i.e., psychosocial vs behavioural, diversity of digital health interventions).

PLEASE COMMENT

Q 4 Is the title appropriate, concise, attractive?

I think the title could be clearer in describing the study (e.g., Effectiveness of an informative video on reducing perceptions of fear, loneliness, and public stigma related to COVID-19: an RCT).

Q 5 Are the keywords appropriate?

Yes

Q 6 Is the English language of sufficient quality?

Yes

Q 7 Is the quality of the figures and tables satisfactory?

Yes.

Q 8 Does the reference list cover the relevant literature adequately and in an unbiased manner?)

Yes

QUALITY ASSESSMENT

Q 9 Originality

Q 10 Rigor

Q 11 Significance to the field

Q 12 Interest to a general audience

Q 13 Quality of the writing

Q 14 Overall scientific quality of the study

REVISION LEVEL

Q 15 Please take a decision based on your comments:

Major revisions.